# Preventing Forklift Front-End Failures: Predicting the Weight Centers of Heavy Objects, Remaining Useful Life Prediction under Abnormal Conditions, and Failure Diagnosis Based on Alarm Rules

**DOI:** 10.3390/s23187706

**Published:** 2023-09-06

**Authors:** Jeong-Geun Lee, Yun-Sang Kim, Jang Hyun Lee

**Affiliations:** 1Department of Smart Digital Engineering, INHA University, Incheon 22212, Republic of Korea; jeonggeun2.lee@doosan.com; 2Doosan Industrial Vehicle, Incheon 22503, Republic of Korea; yunsang.kim@doosan.com; 3Department of Naval Architecture and Ocean Engineering, INHA University, Incheon 22212, Republic of Korea

**Keywords:** PHM, CBM, diagnosis, lightGBM, random forest, contextual diagnosis, RUL, forklift

## Abstract

This paper addresses the critical challenge of preventing front-end failures in forklifts by addressing the center of gravity, accurate prediction of the remaining useful life (RUL), and efficient fault diagnosis through alarm rules. The study’s significance lies in offering a comprehensive approach to enhancing forklift operational reliability. To achieve this goal, acceleration signals from the forklift’s front-end were collected and processed. Time-domain statistical features were extracted from one-second windows, subsequently refined through an exponentially weighted moving average to mitigate noise. Data augmentation techniques, including AWGN and LSTM autoencoders, were employed. Based on the augmented data, random forest and lightGBM models were used to develop classification models for the weight centers of heavy objects carried by a forklift. Additionally, contextual diagnosis was performed by applying exponentially weighted moving averages to the classification probabilities of the machine learning models. The results indicated that the random forest achieved an accuracy of 0.9563, while lightGBM achieved an accuracy of 0.9566. The acceleration data were collected through experiments to predict forklift failure and RUL, particularly due to repeated forklift use when the centers of heavy objects carried by the forklift were skewed to the right. Time-domain statistical features of the acceleration signals were extracted and used as variables by applying a 20 s window. Subsequently, logistic regression and random forest models were employed to classify the failure stages of the forklifts. The F1 scores (macro) obtained were 0.9790 and 0.9220 for logistic regression and random forest, respectively. Moreover, random forest probabilities for each stage were combined and averaged to generate a degradation curve and determine the failure threshold. The coefficient of the exponential function was calculated using the least squares method on the degradation curve, and an RUL prediction model was developed to predict the failure point. Furthermore, the SHAP algorithm was utilized to identify significant features for classifying the stages. Fault diagnosis using alarm rules was conducted by establishing a threshold derived from the significant features within the normal stage.

## 1. Introduction

This paper focuses on the challenge of mitigating durability degradation and accurately predicting failures in forklifts operating under demanding conditions. As classified by the American Society of Mechanical Engineers (ASME), forklifts, powered vehicles used for various material handling tasks, often encounter durability and performance issues while maneuvering heavy loads across logistics warehouses, construction sites, factories, and similar environments. These issues escalate service and maintenance costs, introduce safety hazards, and potentially endanger human lives. To address these challenges, the integration of prognostics and health management (PHM) technologies become essential. PHM, a comprehensive framework leveraging sensors, aims to assess system health, diagnose anomalies, and predict remaining useful life [1]. By encompassing condition monitoring, assessment, fault diagnosis, and prediction, PHM optimizes decision making for condition-based maintenance [2]. It necessitates the application of a systematic methodology to select appropriate feature engineering techniques and algorithms tailored to the specific context [3]. Notably, the study by Meng et al. [4] presented a comprehensive overview of research trends in PHM, particularly focusing on lithium-ion batteries. Categorizing prediction approaches into physics-based, data-driven, and hybrid categories, their work provides valuable insights into the diverse avenues within the field.

Recognizing the limitations of traditional reliability analysis, which relies on mean time to failure data and failure probability distributions, contemporary efforts are directed towards harnessing sensor-based PHM technologies to overcome these challenges. Previous studies in reliability and fault diagnosis have predominantly focused on statistical analyses under average load conditions or known failure scenarios [5,6,7,8,9]. However, the advent of the internet of things (IoT) is driving the transition to sensor-based PHM in a variety of domains as it evolves into a new phase of asset management [10]. Physics-based PHMs formalize the complex behavior of equipment and systems to help diagnose and predict failures when mechanical systems exceed predefined physical thresholds. In contrast, sensor-based methodologies have attracted considerable attention due to their ability to proficiently cope with complex troubleshooting scenarios due to their powerful representation and automated feature learning capabilities [11]. However, sensor-based PHMs face several challenges. Data collected from multiple sensors are prone to noise during acquisition and transmission. Since each type of failure produces different failure signals, signals and failures may not correspond exactly one-to-one in general [12]. The limitations of directly monitoring raw signals in deep-learning-based PHM are recognized, necessitating a transformation process termed feature engineering [12,13]. This process encompasses noise filtration, statistical feature extraction, frequency conversion, and context-dependent conditional reduction, thereby enhancing the data’s meaningfulness and applicability.

Within the data-driven and deep-learning-based PHM framework, considerable research is dedicated to fault diagnosis and the prediction of remaining useful life. While PHM technologies aim for high fault detection accuracy and predictive capabilities, the practicality of models is also contingent on achieving a lightweight design. Ding et al. [14] introduced a lightweight multiscale convolutional network tailored for bearing fault diagnosis in edge computing scenarios, specifically targeting train bogie bearings. To complement model-based lightweighting efforts, feature-based lightweighting is pursued through dimension reduction. Lee et al. [3] emphasized the importance of effective feature engineering and data dimensionality reduction in PHM.

Addressing the challenge of imbalanced data situations, Zhang et al. [15] proposed an integrated multi-task intelligent bearing fault diagnosis method, leveraging representation learning under unbalanced sample conditions. This approach is particularly relevant given the common scenario of skewed sample ratios between fault and normal data. Furthermore, the scarcity of fault data arising in distinct environments necessitates innovative solutions. To overcome this data scarcity challenge, techniques such as data augmentation via generative adversarial networks (GANs) [16] and sensor signal transformation [17] have been advanced. Numerous studies have contributed comprehensive insights into remaining useful life (RUL) prediction within the PHM process [18,19,20,21,22,23,24,25,26,27,28]. RUL prediction encompasses four pivotal technical stages: data collection, construction of health indices, health state segmentation, and RUL prediction [18]. These stages synergistically form a comprehensive framework for accurate prediction.

The Industrial Truck Association (ITA) classifies forklifts into eight distinct product families, ranging from Class I to Class VIII. These classifications are based on specific usage characteristics and structural differentiations. For the purpose of this investigation, a PHM study was undertaken, focusing on the front-end structure responsible for lifting heavy loads. This structural component serves as the core element within ITA Class I-type electric counterbalance forklifts. Durability evaluation of forklifts typically involves testing under average loads and standardized conditions, conforming to established standards such as ISO 3691-1, ANSI/ITSDF B56.1, EN 1726-1, and ASME B56.1. However, the applicability of these conventional statistical approaches can be limited in non-standard operational settings where anomalies can arise. In real-world work environments, forklifts often encounter abnormal loading conditions that can result in rollovers or structural failures. Moreover, when latent strength deficiencies accumulate during assembly, the risk of structural failure becomes significantly elevated.

Among the various safety concerns linked with forklifts, the most significant hazard is the occurrence of vehicle rollovers. These incidents commonly occur when maneuvering heavy objects that exceed the forklift’s designated capacity or when handling unbalanced or improperly centered loads. Furthermore, the repetitive use of forklifts while carrying unbalanced heavy loads can lead to performance degradation over time. This cumulative deterioration can eventually culminate in safety accidents. This particular failure mode operates outside the established standard and stands as an outlier load condition. It resides beyond the boundaries of average reliability ranges and is, therefore, not explicitly accounted for in typical design considerations. Paradoxically, these abnormal failures frequently occur within real-world operational environments. This inherent discrepancy necessitates the development of abnormal failure prevention technologies. These technologies should extend beyond the confines of existing reliability-based lifetime management approaches. Specifically, there is a need for a PHM technique that leverages machine learning and deep learning methodologies, utilizing sensor data to detect changes in forklift operational states. This enables the accurate classification and prediction of impending failures. Consequently, this study introduces an effective PHM approach designed to mitigate safety accidents stemming from abnormal forklift usage conditions. By addressing these anomalies, the aim is to enhance forklift durability, thereby curbing maintenance costs and fostering a value chain that prioritizes accident prevention.

This paper provides a systematic presentation of forklift failure diagnosis and prediction, aiming to ensure the reliability of the forklift front-end, as illustrated in Figure 1. The following overview outlines the content of each section.

In Section 2, vibration data were collected under weight-imbalance conditions, and a subsequent feature engineering process was executed to facilitate a comprehensive classification of the centers of heavy objects. The primary objective of this classification study was to diagnose weight imbalance, a critical factor influencing forklift durability. By addressing this imbalance, the study aimed to proactively prevent factors leading to durability degradation in forklifts, thereby minimizing the risk of vehicle rollover.Continuing in Section 3, vibration and sound data were collected from forklifts operating under abnormal weight-imbalance conditions. Employing another feature engineering process alongside health stage classification, this section further delves into the realm of anomaly detection and classification.Section 4 is devoted to conducting remaining useful life (RUL) analysis and an alarm-rule-based fault diagnosis, serving the overarching goal of enhancing operational maintenance. The RUL prediction was based on the probability model established in Section 3, enabling accurate predictions. Additionally, significant features were extracted from both classifiers and features previously generated in Section 3. This information was then employed for an early front-end failure diagnosis through an alarm-rule-based approach.Lastly, Section 5 encapsulates the findings and conclusions, summarizing the contributions derived from the investigation.

**Figure 1 sensors-23-07706-f001:**
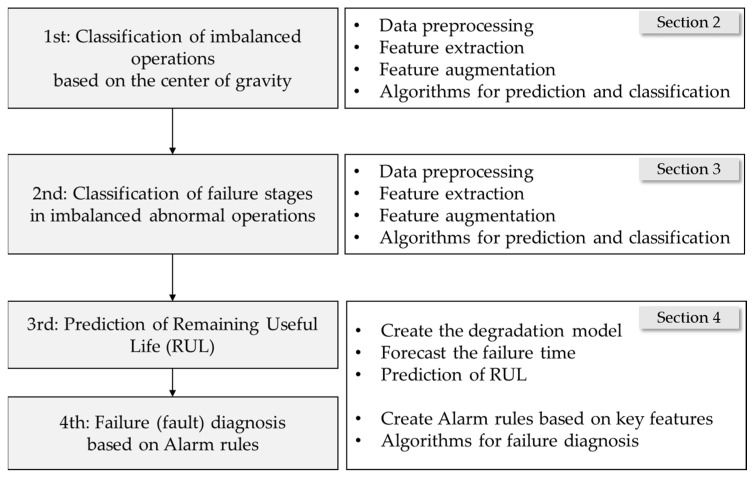
Outline of procedure used to predict the failure of forklift front-end.

## 2. Diagnosing and Classifying the Weight Center of Heavy Objects Carried by Forklifts

This section addresses the diagnosis of weight imbalances, which have a direct impact on the overall durability of forklifts. To achieve this, a structured approach was followed that included several essential steps: data preprocessing, feature engineering, data augmentation, and careful selection and evaluation of appropriate machine learning models. The overall goal was to develop a comprehensive method to detect the center of gravity, as shown in Figure 2. 

### 2.1. Experimental Data Acquisition and Feature Engineering

Vibration (acceleration) data were acquired from the forklift’s front-end structure and presented a process to diagnose and classify the weight center of heavy objects carried by forklifts. The front-end structure of the forklift consists of a mast, backrest, carriage, and forks, as shown in Figure 3, and the acceleration signals were measured from the outer beam of the mast.

In the measurement experiment for data acquisition, the weight center of heavy objects carried by the forklift was measured in three configurations: center, left, and right. The condition segments of the dataset were classified and organized into center, left, and right according to each center of gravity condition. Two embedded devices (one on the left and one on the right) were attached to the front-end structure of the forklift truck to measure the vibration acceleration in three axes (x, y, z) (sampling rate 500 Hz), as shown in Figure 4. Considering the load conditions under which the forklift operates, the operating environments of the two datasets (datasets 1 and 2) were simulated in the experiments while maintaining a state that included ground noise. In dataset 1 (only driving mode), the vehicle was loaded with 3200 kg of weight and traveled 80 m at maximum speed, as shown on the left in Figure 5. All measurements were taken for approximately 20 min for center, left, and right condition segments to eliminate data imbalances within condition segments. In dataset 2 (complex mode), the forklift made a round trip of 80 m, as shown on the right in Figure 5, and added lifting, lowering, and back-and-forth tilting tasks at the end of the trip. In dataset 2, approximately 32 min of data were acquired, which is 12 min longer than in dataset 1. Similarly, approximately 32 min of data were acquired for the center, left, and right condition segments to eliminate the data imbalance.

Six acceleration signals (2 sensors × (x, y, z accelerations)) were included in Data sets 1 and 2. In Dataset 2, for one acceleration signal, 960,000 feature vectors were collected at 500 (Hz) × 32 (min) × 60 (s/min). Given the large data size, which could lead to inefficient analysis, this study referred to previous studies [27,28] to handle the data. Eight features were extracted from each window at one-second intervals: min, max, peak to peak, mean (abs), rms (root mean square), variance, kurtosis, and skewness. In addition, four features were added by combining the max, rms, and mean (abs) features: crest factor, shape factor, impulse factor, and margin factor, as listed in Table 1.

Through the above process, 12 features were extracted, and the measured data were compressed for efficient analysis. The data compression process transformed the dimensionality of the data from 960,000 × 6 × 1 to 1920 × 6 × 12. In this way, the number of feature vectors in the data was reduced, but the number of features was increased twelvefold, resulting in 72 features. The aim was to enable effective data processing and facilitate the diagnosis of the weight center of heavy objects at a one-second interval. Furthermore, the data range of the feature vectors was scaled from 0 to 1 using min–max normalization. Furthermore, an exponentially weighted moving average (EWMA) with a window size of 2 to 3 s was used to smooth the noise signals of the generated features. This approach helped to minimize the noise and outliers in the feature data. Moving averages average out the effects of past data, and exponentially weighted moving averages have the advantage of exponentially attenuating these effects. Therefore, they are used for tasks such as time series forecasting and noise reduction [29,30]. The exponentially weighted moving average is used in a way that adjusts the value of alpha (α) based on the window size, as shown in Equation (1). As described in Equation (3), this method smooths out noise in the feature vector to minimize outliers while also reducing the influence of past vectors.
(1)$$\alpha =\frac{2}{window\;size+1},\;\;\;\;window\;size\geq 1$$
(2)$$y_{0}=x_{0}$$
(3)$$y_{t}=\frac{x_{t}+(1-\alpha )x_{t-1}+(1-\alpha {)}^{2}x_{t-2}+\ldots +(1-\alpha {)}^{t}x_{0}}{1+(1-\alpha )+(1-\alpha {)}^{2}+\ldots +(1-\alpha {)}^{t}}$$

The parameters employed within Equations (1) through (3) are listed below:

α: the weight coefficient of exponentially moving average.window size: the number of data points used by EWMA.$y_{t}$: the exponentially weighted moving average value at the current time *t*.$x_{t}$: the input data value at the current time *t*.$x_{t-1},\;x_{t-2},\ldots ,x_{0}$: the input data values at past time points.$(1-\alpha {)}^{t},\;(1-\alpha {)}^{t-1},\ldots ,\;1$: the weights of past time points.$1+(1-\alpha )+(1-\alpha {)}^{2}+\ldots +(1-\alpha {)}^{t}$: the sum of weights.

The exponentially weighted moving average was applied to take advantage of its ability to minimize outliers in the feature vector. Figure 6 shows the ‘min’ feature extracted from the x-acceleration signal using the EWMA technique. Table 2 and Table 3 contrast features extracted from the dataset before and after the application of EWMA. They showcase the initial data state alongside the effects of EWMA, including smoothing and value adjustments.

Through the previous feature engineering process, as shown in Table 4, 3699 feature vectors were generated from dataset 1 (only driving) in approximately 20 min, and 5755 feature vectors were derived from dataset 2 (complex mode), measured for approximately 32 min. In total, 9454 datasets were obtained, which were further divided into training and test datasets at a 7:3 ratio. The training and test datasets comprised 6617 and 2837 samples, respectively, with each feature vector containing 72 features. An attempt was made to use the training dataset to develop machine learning classifier models and check the performance of the machine learning models on the test dataset.

The number of feature vectors in the dataset was also augmented to minimize overfitting during the training process. Data augmentation was performed using additive white Gaussian noise (AWGN) and long short-term memory autoencoder (LSTM AE), which expanded the training dataset to a maximum of 19,851 samples (Table 5).

AWGN was applied by referring to prior studies [31], and the target signal-to-noise ratio (SNR) was set to 20 dB. An additional dataset could be generated by mixing noise with the original data, as shown in Figure 7. AWGN is a method of adding noise with a Gaussian distribution to the input or output signal of a system. SNR serves as a scale that quantifies the ratio between the signal’s strength and the noise level. A higher SNR value corresponds to a more robust signal, reducing the relative impact of noise. AWGN based on SNR can be expressed as follows [31]:(4)$$N\left(t\right)=A\sqrt{\frac{10^{-SNR/10}}{2}}\cdot w\left(t\right)$$

$N\left(t\right)$: the noise of the signal.A: the magnitude of the input or output signal.SNR/10: the standard deviation *σ* of the noise, calculated from the SNR value.$w\left(t\right)$: the white noise, which follows a Gaussian distribution with a mean of 0 and a variance of 1.

**Figure 7 sensors-23-07706-f007:**
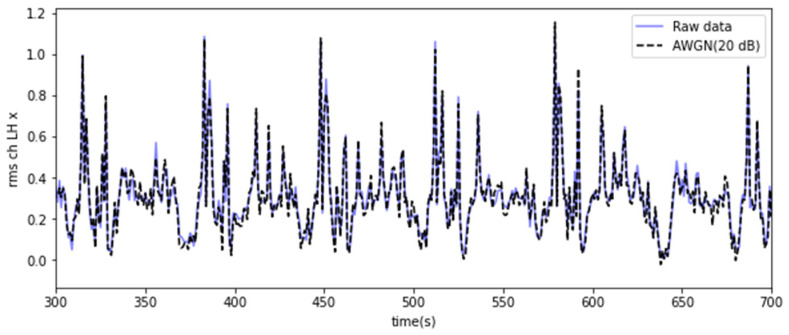
Results of noise mixed augmentation using AWGN.

The autoencoder is a neural network that can use unlabeled training data to learn a code that efficiently represents the input data. This type of coding is useful for dimensionality reduction because it typically has much lower dimensionality than the input. In particular, it works as a powerful feature extractor that can be used for the unsupervised pre-training of deep neural networks. An autoencoder consists of an encoder that converts the input to an internal representation and a decoder that converts the internal representation back to output [32]. The output result is called reconstruction because the autoencoder reconstructs the input. This study used the mean square error (MSE) as the reconstruction loss in training. LSTM, an artificial recurrent neural network, was designed to address the vanishing gradients in traditional recurrent neural networks (RNNs) [33]. As the number of hidden layers in a neural network and the number of nodes in each layer increase, the last layer is trained while the initial layer is not trained.

This long-term dependency problem arises from the vanishing gradient problem, where the gradients tend to converge to zero during the gradient propagation process, particularly when the data length increases during the training phase of RNNs [34,35]. On the other hand, unlike traditional RNNs, LSTM can effectively overcome the vanishing gradient problem by incorporating long- and short-term state values in the learning process, enabling successful learning even with long training durations [36,37]. In this study, an LSTM autoencoder consisting of two LSTM layers was implemented because time series data were used, as shown in Figure 8. Each layer is used as an encoder and decoder [38]. Furthermore, the repeat vector was used in the decoder part to restore the compressed representation to the original input sequence. The repeat vector function repeats the compressed latent space representation to produce a representation that matches the sequence length. This allows the decoder to use the compressed representation multiple times to reconstruct the original input sequence. Using the LSTM autoencoder, the original feature vectors were trained as input data, and the output vectors were used as the augmentation dataset. The output dataset was generated and replicated to minimize the MSE, resulting in a dataset with similar characteristics and patterns to the input dataset, as shown in Figure 9. The equations and parameter descriptions for the LSTM autoencoder are as follows [32,33].
(5)$$h_{t}=LSTM_{encoder}\left(x_{t},h_{t-1}\right)$$
(6)$$z_{t}=f\left(W_{z}h_{t}+b_{z}\right)$$
(7)$$h_{t}^{\prime }=LSTM_{decoder}\left(z_{t},h_{t-1}^{\prime }\right)$$
(8)$$\hat{x_{t}}=f\left(W_{x}h_{t}^{\prime }+b_{x}\right)$$
(9)$$MSE=\frac{1}{n}\sum_{i=1}^{n}{\left(x_{i}-\hat{x_{i}}\right)}^{2}$$

$x_{t}$: the input time series data.$z_{t}$: the output (latent variable) of the encoder.$\hat{x_{t}}$: the output (reconstructed time series data) of the decoder.$h_{t}$ and $h_{t}^{\prime }$: the hidden states of the LSTM.$W_{z}$, $b_{z}$, $W_{x}$, and $b_{x}$: the learnable parameters (weights and biases) of the model.f: the activation function, typically sigmoid or tanh function.MSE: the mean squared error, loss function.$x_{i}$: the *i*th element of the input data.$\hat{x_{i}}$: the *i*th element of the model’s prediction (reconstructed data).n: the number of elements in the input data.

**Figure 8 sensors-23-07706-f008:**
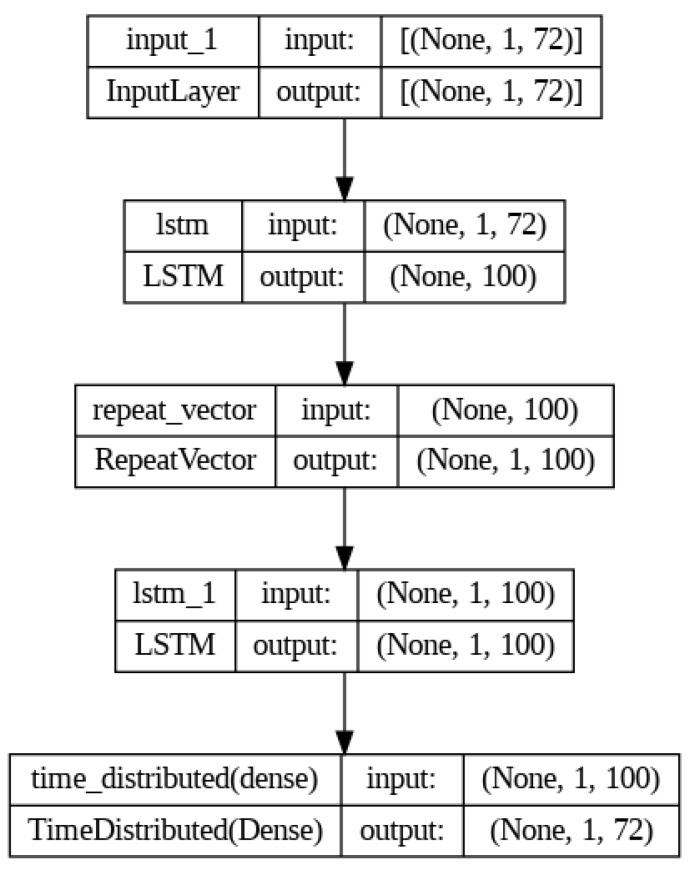
Structure of the LSTM autoencoder layers.

**Figure 9 sensors-23-07706-f009:**
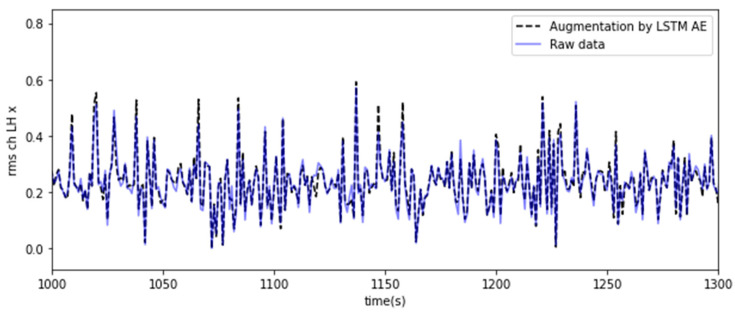
Result of feature augmentation using the LSTM autoencoder.

### 2.2. Result of Classification

To compare the accuracy of failure prediction, the selected classification algorithms were random forest [39] and LightGBM [40]. To enhance their performance, the ‘Bayesian optimization’ method was employed for hyperparameter tuning. Random forest, an ensemble technique, is rooted in decision trees and serves as a classifier. Decision trees build tree-like models based on input variables, efficiently growing the tree by identifying optimal splitting rules at each branch. However, the vulnerability of a single decision tree to overfitting can hinder its ability to generalize well. To address this concern, the random forest algorithm is applied to alleviate overfitting concerns. A decision tree, by itself, operates as a tree algorithm for data classification or prediction. It navigates the classification or prediction process by creating a tree structure grounded in the data, partitioning it into multiple child nodes through evaluations of specific conditions at each node. The criteria of these conditions are typically determined by metrics such as information gain (IG) or the Gini index (Gini). These metrics measure the impurity of class distribution at each node, selecting a splitting criterion that minimizes the difference in impurity before and after the split [39,40].
(10)$$\mathrm{Information}\;\mathrm{gain}:\;IG\left(D_{p},f\right)=I\left(D_{p}\right)-\sum_{j=1}^{m}\frac{N_{j}}{N_{p}}I\left(D_{j}\right)$$
(11)$$\mathrm{Gini}\;\mathrm{index}:\;Gini\left(D_{p}\right)=1-\sum_{k=1}^{K}p_{k}^{2}$$

$D_{p}$: the data of the parent node.$D_{j}$: the data of the jth child node.f: the splitting criterion variable.m: the number of child nodes generated after splitting.$N_{p}$: the number of data points in the parent node.$N_{j}$: the number of data points in the jth child node.K: the number of classes.$p_{k}$: the ratio of the kth class.

Random forest stands out as a notable ensemble learning technique, leveraging the power of decision trees. The methodology of random forest is structured around the collaborative efforts of multiple decision trees, which are subsequently aggregated to yield prediction results. The workflow of random forest unfolds as follows:Bootstrap sample creation: The process starts by randomly selecting a subset of the input data to create what is known as a bootstrap sample. This sample comprises a distinct dataset consisting of data instances randomly extracted from the original input dataset.Multiple decision tree generation: Next, numerous decision trees are generated, each stemming from a bootstrap sample. These decision trees come into existence with a random element, ensuring their diversity and independence.Data prediction by decision trees: Each of the generated decision trees is then utilized to predict the input data. This prediction process is carried out individually for all the decision trees in the ensemble.Aggregation of predictions: The prediction results obtained from the individual decision trees are aggregated. This aggregation can take the form of averaging the predictions or adopting a majority voting approach, depending on the task. The aggregated outcome serves as the foundation for the final predictions made by the random forest model.

By following these steps, random forest harnesses the collective insights of multiple decision trees, effectively enhancing prediction accuracy and generalization capabilities. Random forest addresses overfitting by generating multiple models from different data subsets. This diversification improves robustness against noise and uncertainties. The adjustment of the optimal number of decision trees and the splitting criteria can be performed through hyperparameter tuning. Typically, hyperparameters such as the splitting criteria of decision trees and the tree depth are set. The prediction function of random forest is as follow, where T is the number of generated decision trees and $f_{t}\left(x\right)$ represents the prediction function of the tth decision tree.
(12)$$f\left(x\right)=\frac{1}{T}\sum_{t=1}^{T}f_{t}\left(x\right)$$

LightGBM is a machine learning model based on the gradient boosting decision tree (GBDT) algorithm. Gradient boosting works by improving the prediction model as new models compensate for the errors of the previous model. Therefore, multiple decision trees can be combined to develop a more robust model that minimizes overfitting. The working mechanism of lightGBM is similar to the conventional GBDT algorithm, but it utilizes the leaf-wise approach during the splitting process (Figure 10). This approach allows lightGBM to produce more unbalanced trees than the traditional level-wise approach, resulting in improved predictive performance. Furthermore, lightGBM includes the feature to perform splitting using only a subset of the data, ensuring faster processing speed for large-scale datasets. However, lightGBM may result in deeper trees, depending on their leaf-wise characteristics and hyperparameter settings, which may lead to deeper trees [41]. While this can improve the prediction accuracy of the training data, it may result in lower accuracy when predicting new data because of the overfitting problem. In the case of lightGBM, the aim was to minimize the overfitting problem through the feature augmentation conducted previously. The objective function and parameter descriptions for LightGBM are as follows [42]:(13)$$Obj\left(\Theta \right)=\sum_{i}l\left(y_{i},y_{i}\hat{y_{i}^{\left(t-1\right)}}+f_{t}\left(x_{i}\right)\right)+\sum_{t}\Omega \left(f_{t}\right)$$
(14)$$\Omega \left(f_{t}\right)=\gamma T+\frac{1}{2}\lambda {\left|w\right|}^{2}$$

$l\left(y_{i},\hat{y_{i}^{\left(t-1\right)}}+f_{t}\left(x_{i}\right)\right)$: the loss functions used in the objective function.$y_{i}$: the actual value for the ith data.$\hat{y_{i}^{\left(t-1\right)}}$: the prediction from the previous time step (*t* − 1).$f_{t}\left(x_{i}\right)$: the prediction of the tth tree for the ith data point $x_{i}$.$\Omega \left(f_{t}\right)$: a term used to regulate the complexity of the tree.T: the number of leaf nodes in the tree.γ: the cost parameter associated with the number of leaf nodes.λ: a coefficient that regulates the weight of leaf nodes.w: the weights assigned to the tree nodes.

**Figure 10 sensors-23-07706-f010:**
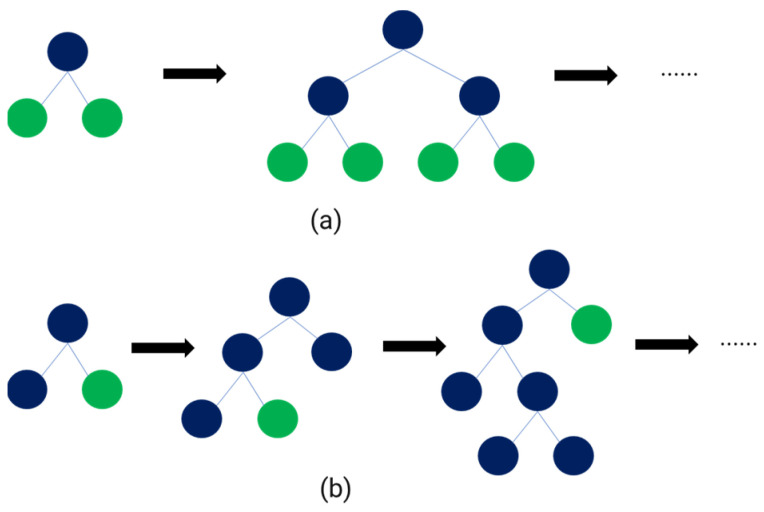
Two kinds of tree growth: (**a**) level-wise growth, (**b**) leaf-wise growth.

Bayesian optimization is a method for finding the optimal solution that maximizes an arbitrary objective function. This optimization technique can be applied to any function for which observations can be obtained and is particularly useful for optimizing black-box functions with high cost and unknown shapes [43]. Therefore, Bayesian optimization is used mainly as a hyperparameter optimization method for machine learning models, taking advantage of the characteristics of such optimization techniques [44]. The optimized hyperparameters were derived through Bayesian optimization, as shown in Table 6. In Bayesian optimization, the aim is to identify the hyperparameter combination $x^{*}$ that minimizes or maximizes the objective function $f\left(x\right)$. The objective function typically takes a form similar to Equation (15). $\eta \left(x\right)$ represents the actual value of the objective function for the hyperparameter combination x. $f\left(x\right)\;$is defined as the sum of the actual objective function value $\eta \left(x\right)$ and the noise $\epsilon \left(x\right)$. Bayesian optimization involves experimentation with various hyperparameter combinations while modeling both the genuine objective function value $\eta \left(x\right)$ and the accompanying noise $\epsilon \left(x\right)$. Through iterative processes, the next hyperparameter combination to explore is forecast, advancing the optimization process.
(15)$$f\left(x\right)=\eta \left(x\right)+\epsilon \left(x\right)$$

Because forklifts move continuously, vibration data has the characteristics of time series data. The time series data and the state changes in the condition segment (center of heavy objects carried by forklift) do not depend on the state at a single point in time but on the past values. Therefore, the probability of classifying the conditioning segment was the EWMA to diagnose the conditioning segment contextually using the moving average instead of diagnosing the conditioning segment only by the probability at that time, as shown in Table 7. Contextual diagnosis in machine learning is a technique to diagnose by considering the context of the given data [45]. This provides a more profound understanding than simply analyzing and predicting data patterns. It simply considers the context of the data before and after, rather than individual data points, to help make an accurate diagnosis. In addition, it is used effectively for outlier detection in time series [46] and partial data [47]. This study attempted to minimize the effect of noise, such as outliers, using the exponentially weighted moving average for contextual diagnosis. In applying contextual diagnosis, this study examined the effects of the window size on the moving average. Figure 11 presents the learning and prediction process flow. 

As a result of contextual diagnosis through the exponentially weighted moving average, the classification probability of each condition segment predicted by machine learning changes, as listed in Table 7. Forklifts carry and transport unbalanced heavy objects during continuous movement or operation, and the centers of heavy objects do not fluctuate on a one-second basis. Therefore, a two to three-second window was used to calculate the moving average probability. As a result, when the condition segment was diagnosed as “center”, applying a moving average to the probabilities in certain outlier segments diagnosed as “left” or “right” would result in lower values influenced by past data. Figure 12 and Figure 13 present these probabilities as graphs as a function of time. From the observed results, the centers of heavy objects carried by the forklift were diagnosed more accurately by the generated classifier.

Because the data imbalance was minimized, the performance was compared using the accuracy score of 24 classifiers, including random forest and lightGBM (Table 8). First, the accuracy increased in all cases as the window size of the exponentially weighted moving average for smoothing the feature vector time series data increased and the alpha value decreased. The classification probabilities of the condition segment predicted by machine learning were subjected to the moving average to achieve a contextual diagnosis. In all cases, the classification accuracy of the condition segment increased gradually as the size of the moving average window increased, and the alpha decreased. On the other hand, although the data augmentation minimized the overfitting of the model, it resulted in the same or slightly lower accuracy.

Applying moving average to the probabilities of the lightGBM model resulted in an overall increase in the scores (cases 13–24). This trend can be seen in the probability graphs diagnosing each condition segment in Figure 12 and Figure 13. However, lightGBM, with its leaf-wise growth strategy, tended to overfit with increasing tree depth and often has probabilities highly skewed towards 0 or 1. Therefore, it was difficult to observe performance improvement in the combination of the augmented dataset and contextual diagnosis for the model (cases 13, 16, 19, and 22).

Compared to lightGBM, random forest exhibited relatively less overfitting and showed gradual fluctuations in probabilities corresponding to changes in the time series. This trend can be observed in Figure 13. The diagnosis of the condition segments was not sigmoidal but rather smooth and gradual because the probability was calculated by averaging the voting values of multiple randomly generated decision trees. These characteristics were expressed in classifiers utilizing the augmented dataset with AWGN and LSTM autoencoders. When applying AWGN and LSTM autoencoders to the dataset and contextual diagnosis in cases 10–12, the application of the probability moving average resulted in a higher score of 0.0331 (3.31%) compared to cases 1–3. On the other hand, the random forest score was lower than lightGBM in all cases when the moving average was not applied (without a contextual diagnosis).

By applying machine learning and contextual diagnosis, the diagnosis of the centers of heavy objects carried by forklifts was performed by the condition segment during the process of lifting, tilting, and moving heavy objects on uneven ground using a forklift. As a result, the random forest model (case 12) achieved a maximum accuracy of 0.9563, while the lightGBM model (case 24) achieved a maximum accuracy of 0.9566.

## 3. Abnormal Lifting Weight Stage Classification

In Section 3, vibration and sound data were acquired from forklifts operating under repeated abnormal weight lifting conditions. In addition, feature engineering and health stage classification of the data were performed following a systematic PHM procedure as depicted in Figure 14. 

### 3.1. Experimental Data Acquisition and Feature Engineering

Safety accidents and failure situations were induced by continuous lifting of unbalanced heavy objects in a laboratory environment. Data measurements and condition diagnoses were conducted for these situations. During the data measurement process, the forklift repeatedly lifted and lowered an unbalanced load of 1500 kg to the right at a consistent speed every 20 s for five hours. The forklift remained stationary throughout this process and did not perform any driving activity. In addition, four three-axis (x, y, z) acceleration sensors were mounted on the left and right sides of the front-end structure to collect vibration data (Figure 15). Repeated acceleration tests were performed until a failure condition occurred; the forklift swayed, and a loud noise was generated in the structure. In the actual operating environment, it is difficult to diagnose the condition and predict the lifespan by sound because of ambient noise. In this study, a microphone (sampling rate 51.2 kHz) was installed at the driver’s position to eliminate ambient noise in an anechoic chamber environment, and labeling was performed using sound data. At the same time, the noise generated was measured by the forklift. These sound data were used to classify and label the failure stages into three stages: normal, failure entry, and failure.

The dataset obtained in the experiment contains 12 acceleration signals (four sensors × (x, y, z accelerations)). For each signal of one sensor, 512 (Hz) × 300 (min) × 60 (s/min) acceleration data were measured, resulting in a total of 9,216,000 data points. In addition, 921,600,000 (51,200 Hz × 300 min × 60 s/min) sound data were collected. As shown in Figure 16, the plotted sound signal obtained in the time domain made it difficult to track the state changes. By observing the frequency changes over time using the short-time Fourier transform (STFT), it was possible to determine if the forklift was operating over time. On the other hand, state changes and detailed differences were difficult to compare.

The data size was large, and it was difficult to observe the distinctive state changes when analyzing the raw signals. Twelve features were extracted from the time-domain signal, including min, max, peak to peak, mean (abs), RMS, variance, kurtosis, skewness, crest factor, shape factor, impulse factor, and margin factor for every 20 s window to compress the data and solve these difficulties. Based on this, the dimensions of the acceleration signal and sound signal data were reduced to 900 × 12 × 12 and 900 × 1 × 12, respectively. As reported in Section 2, an exponentially weighted moving average (alpha 0.5) was applied to the reduced dataset to minimize the noise signal in the data. After feature engineering, the CART (classification and regression tree) algorithm [40] was used to classify the failure stages over time using sound features. The CART algorithm is a decision tree algorithm for classification and regression analysis that evaluates the importance of each variable based on the input data and prioritizes the important variables to produce a decision tree. The CART algorithm was used to derive 12 decision trees for each feature by pruning to prevent overfitting and classify the status into three levels (Figure 17). Significant branching points could be derived from eight of the 12 decision trees, and the failure stage was classified based on the average value of the branching points derived from the eight decision trees (Table 9). Based on the classification results, breakpoint 1 was determined to be at 1.18 h and breakpoint 2 at 1.77 h, which were used to distinguish the labeling of the data (Table 10). Furthermore, a comparison of the recorded forklift experimental video showed that beyond breakpoint 1, the forklift exhibited early failure symptoms (beginning to shake), demonstrating a complete failure state at breakpoint 2. 

The 900 data generated were divided into 630 data for training and 270 data for testing. Given that the dataset is imbalanced, the performance was evaluated using the F1 score during the validation process. The F1 score is one of the metrics that evaluate the accuracy of a model and is calculated as the harmonic mean of precision and recall. When data are unbalanced, the prediction accuracy for a small number of data classifications is often high, but the prediction accuracy for a large number of data classifications is often low. In such situations, relying solely on accuracy may give the impression of high prediction accuracy for a small number of data classifications. On the other hand, the accuracy for most data classifications may be relatively lower, leading to inadequate overall model performance evaluation. The F1 score considers both precision and recall, which makes it more suitable for dealing with imbalanced classification problems [48,49,50]. Therefore, when the data are unbalanced, using the F1 score is a more accurate way to evaluate the model performance. Consequently, the F1 score was used to evaluate the performance in this unbalanced dataset.

### 3.2. Stage Classification Result

In the previous steps, labels were assigned to instances using sound features. Leveraging these labels, logistic regression and random forest classifiers were trained. The dataset was divided into two distinct cases: case 1 utilized solely vibration features, encompassing 144 dimensions; in contrast, case 2 integrated both vibration and sound features, totaling 156 dimensions. While sound features were primarily introduced for labeling purposes, their inclusion in the training prompted additional cases to assess their impact on classifier performance. In both cases, logistic regression and random forest were selected as the machine learning models, with the details of random forest elaborated upon in Section 2.2. For multiclass classification, logistic regression employs the SoftMax function alongside the cross-entropy loss. In this context, the linear discriminant function $h_{k}\left(x\right)$ is defined as follows:(16)$$h_{k}\left(x\right)=w_{k}^{T}x_{i}+b_{k}$$
where $w_{k}$ and $b_{k}$ denotes the weights and bias for each class k, respectively, while $x_{i}$ represents the ith input variable. $p\left(y=j|x_{i}\right)$, the probability of class j, predicted by the SoftMax function is as follows [32]:(17)$$p\left(y=j|x_{i}\right)=\frac{e^{h_{j}\left(x_{i}\right)}}{\sum_{k=1}^{K}e^{h_{k}\left(x_{i}\right)}}$$

The cross-entropy loss function, $J\left(\theta \right),$ is calculated as follows [32]: (18)$$J\left(\theta \right)=-\frac{1}{N}\sum_{i=1}^{N}\sum_{j=1}^{K}y_{ij}\log\hat{y_{ij}}$$

$y_{ij}$: the binary value indicating whether class j is the correct target for the ith data point.$\hat{y_{ij}}$: the predicted probability of class j for the ith input data point.N: the total number of data points.K: the total number of classes.

LightGBM, used in the previous section, is prone to overfitting when the dataset size is small. LightGBM was deemed inappropriate because this dataset consists of 900 feature vectors and was not used in this study. No additional hyperparameter tuning was conducted, and only the default parameters provided by the scikit-learn module in Python were utilized. Based on the validation results of the test data shown in Table 11, the logistic regression classifier achieved an F1 score (macro) of 0.9599 for case 1 and 0.9790 for case 2. In the case of random forest, the F1 scores (macro) for cases 1 and 2 were 0.9116 and 0.9220, respectively. Additionally, Figure 18 depicts a confusion matrix, where the x-axis shows the actual (ground truth) class labels, and the y-axis shows the predicted class labels generated by a model. This confirms that it is possible to classify the forklifts as healthy or aging under repeated unbalanced load weight conditions based on acceleration signals.

## 4. RUL Prediction and Fault Diagnosis with Alarm Rule for Abnormal Lifting

Numerous studies [18,19,20,21,22,23,24,25,26,27,28] have comprehensively explored RUL prediction aligned with fault diagnosis. Zhang et al. [19] introduced an inventive approach by parallelly integrating spatial and temporal features using a hybrid neural network. This model combined a 1D convolutional neural network (CNN) with a bidirectional gated recurrent unit (BiGRU). Furthermore, they assessed the limitations of RUL prediction using CNN, LSTM, and the transformer algorithm on aircraft turbofan engine data. To overcome these limitations, they introduced the integrated multi-head dual sparse self-attention network (IMDSSN), an architecture incorporating the ProbSparse self-attention network (MPSN) and LogSparse self-attention network (MLSN) components [20]. Other strategies in RUL prediction span a broad spectrum. Pham et al. [21] proposed RUL prediction for a methane compressor, employing flow system identification, proportional hazard modeling, and support vector machines. Loutas et al. [22] introduced an ε-support vector machine-based approach for estimating rolling bearing RUL. Gugulothu et al. [23] developed Embed-RUL, addressing differing patterns in embeddings between normal and degraded machines. This technique employs a sequence-to-sequence model based on RNNs to generate embeddings for multivariate time series subsequences. Hinchi et al. [24] introduced a method for bearing RUL prediction centered on a convolutional LSTM network. Niu et al. [25] proposed an RUL prediction technique utilizing a 1D-CNN LSTM network. Jiang et al. [26] presented a time series multiple channel convolutional neural network (TSMC-CNN) integrating CNN with LSTM and attention mechanisms. They further integrated TSMC-CNN with an attention-based long short-term memory (ALSTM) network for bearing RUL prediction. Saidi et al. [28] presented RUL prediction using support vector regression (SVR). These regression-based RUL predictions require well-structured feature vectors and substantial RUL data collection. To address this challenge, several studies have focused on performance enhancement, employing ensemble techniques and a range of model-based approaches, including CNN, LSTM, transformers, attention mechanisms, and SVR. While many works have explored convolutional LSTM networks and sequence-to-sequence models, Ley et al. [18] systematically outlined four technical processes including data collection, health index construction, health stage segmentation, and RUL prediction, with a dedicated focus on RUL. 

Considering the algorithms proposed in the literature and the model of Ley et al. in [18], the present study combines appropriate algorithms in an integrated manner, adapted to the unique demands of abnormal lifting scenarios. This customized integration facilitates the model in capturing latent patterns and interactions arising from unbalanced load conditions, thus leading to enhanced RUL predictions. This section demonstrates real-time active diagnosis using the life model equation, even with limited data, thereby enhancing the practicality of the proposed approaches. As Section 3 discussed the classification of health stages, subsequently, in Section 4, the dataset and classifiers are applied to perform RUL prediction and fault diagnosis using alarm rules, thereby establishing a foundation for condition-based management [51]. This study also introduces real-time active forklift condition diagnosis using the life model equation, even with limited data. Moreover, alarm rules are devised utilizing key features extracted from the health stage classifier and designated as health indices. Figure 19 summarizes the RUL calculation process and the methods used.

### 4.1. Life Prediction Model and RUL Verification

In the case of logistic regression, as shown in Figure 20, the classification probabilities of the classifier were derived as either 0 or 1, making it difficult to track the progressive changes in the state. Therefore, it was not feasible to adopt degradation curves for logistic regression. On the other hand, as shown in Section 3, when the classification probabilities of the entire dataset belonging to each stage were visualized using a random forest classifier, a gradual change in state was observed in the graph (Figure 21). This represents the degradation state of the forklift using classification probabilities, and the probabilities of Stages 1 and 3 were combined and averaged to generate the degradation curve (Figure 22). After analyzing the degradation curve, Stage 3 was reached after approximately 106 min, and the threshold was a probability of 0.7.

The model for RUL was constructed using an exponential function equation, as follows.
(19)$$\;y=a+b\times exp(c\times x_{t})$$

Exponential coefficients were determined using the method of least squares, focusing on data collected within the 30 min prior to the diagnostic time. The least squares method is a statistical tool used in regression analysis to determine model parameters that reduce the discrepancy (error) between observed data and the values expected by the model. The following equations and parameters illustrate the implementation of the least squares method with exponential functions, specifically in Equation (20), for a given dataset.
(20)$$Dataset:\;(\left(x_{1},y_{1}\right),\left(x_{2},y_{2}\right),\ldots ,\left(x_{n},y_{n}\right))$$
(21)$$e_{i}=y_{i}-\left(a+b\times \exp\left(c\times x_{ti}\right)\right)$$
(22)$$E=\sum_{i=1}^{n}e_{i}^{2}=\sum_{i=1}^{n}{\left(y_{i}-\left(a+b\times \exp\left(c\times x_{ti}\right)\right)\right)}^{2}$$

$y_{i}$: the observed value of the dependent variable at a point$x_{ti}$: the value of independent variable $x_{i}$ at a time t.a,b, c: the parameters of the exponential function.$e_{i}$: the residuals associated with the exponential function’s prediction.E: the objective function to compute the sum of squared residuals, which is minimized to determine the values of a,b, and c.

Based on these characteristics, the exponential function continued to change with time, and the time when the y-value of the exponential function reached 0.7 was calculated to analyze the RUL. During the analysis of the RUL, the confidence interval was included by considering ${Y(0.9x}_{t})$ to Y(1.1$x_{t})$ in the pre- and post-prediction time points. The analysis showed that the early data predicted the failure rather early, as shown in Figure 23. The gap between the actual and the predicted RUL decreased gradually as the failure point approached (Figure 24).

### 4.2. Alarm-Rule-Based Fault Diagnosis

The SHAP (Shapley additive explanations) algorithm was used to diagnose faults based on the key features and alarm rules. The goal was to determine which features were critical for state classification using the SHAP algorithm with the previously trained random forest classifier. This analysis allowed the identification of the significant features used for state classification. SHAP is an algorithm used to analyze the contribution of features to the predictions made by machine learning models. It helps to determine the importance of each feature in the model’s predictions. The SHAP algorithm considers all features necessary to explain the model predictions and calculates the impact of each feature on the prediction outcome [52]. Calculating the Shapley value for all feature combinations is computationally expensive. Therefore, the SHAP algorithm uses approximations that can be calculated relatively quickly for tree-based models, such as the random forests or XGBoost [29]. The SHAP algorithm provides various interpretation results, such as feature importance, feature contribution, and feature effect, which help to interpret the model and explain the prediction results reliably [52]. The SHAP algorithm improves the interpretability of machine learning models and plays a vital role in model development and helping users to understand the model prediction results [52].

The features that contribute significantly to health diagnostics were identified using the SHAP algorithm. As shown on the left in Figure 25, the ‘abs.mean ch In LH z’ feature contributes the most to the health diagnosis. A progressive state change, similar to the degradation model graph, could be observed by observing the dispersion of the time series of the corresponding feature (Figure 25, right image). The 2-sigma and 3-sigma values of the ‘abs.mean ch In LH z’ feature were extracted from the distribution of the normal data range in stage 1. These values were then used as the threshold for fault diagnosis. Next, an alarm rule was set to diagnose a failure when the exponentially weighted moving average value of the feature (alpha 0.17) exceeds a threshold. As shown in Figure 25 (right), the 3-sigma threshold can diagnose failure just before failure, and the 2-sigma threshold can predict failure at the point of the precursor symptoms.

## 5. Discussion

This study addresses the critical issue of preventing front-end failures in forklifts by predicting the center of gravity. The study begins by emphasizing the fault diagnosis of front-end failures in forklifts and the significance of accurately predicting the gravity center. In this pursuit, the study acquired acceleration signals during lifting, tilting, and moving heavy objects by the forklift on uneven ground and in various operating environments. These acceleration signals were captured from the outer beams on both the left and right sides of the forklift’s front-end. To ensure effective processing and feature determination, time-domain statistical features were extracted and established as variables by applying a window at one-second intervals. To mitigate potential overfitting, the dataset was augmented with AWGN and LSTM autoencoders. Following these data enhancements, classifiers were used to accurately categorize the center of objects being transported by forklifts during their driving and working phases. This classification task was accomplished using the robust capabilities of the random forest and lightGBM models. The random forest model and lightGBM model were able to predict the center of gravity with an accuracy of 0.9563 and 0.9566, respectively. During the prediction of the gravity center, an exponentially weighted moving average was conducted to smooth out the noise of the features. As a result of applying this moving average, the dataset’s outliers were reduced without necessitating complex noise filtering methods. Moreover, an observation was made: as the window size for the exponentially weighted moving average was increased, the accuracy of the machine learning models also improved. This suggests that a larger window captured more accurate data trends, enhancing predictive performance. By using data augmentation, overfitting was minimized, while maintaining similar or slightly lower accuracy scores. In the lightGBM model, the implementation of a moving average on the classification probabilities showed a tendency to improve the scores. In contrast, lightGBM tended to overfit with increased tree depth, resulting in biased classification probabilities. This made it difficult to improve performance when combining augmented datasets with contextual diagnosis. However, the random forest model showed less overfitting compared to lightGBM and showed gradual changes in classification probabilities over time. These trends were observed in the augmented dataset using the AWGN and LSTM autoencoders. Applying the augmented dataset and contextual diagnosis, along with the moving average to the classification probabilities, resulted in an average score improvement of 3.31%. Notably, the random forest score was lower than the lightGBM score without the moving average.

After predicting the center of gravity, the paper presented a procedure for forecasting the RUL during abnormal operational scenarios and diagnosing failures through the application of alarm rules. To forecast RUL during repetitive unbalanced load conditions, statistical features were extracted from acceleration data using a 20 s window. This data collection took place in an anechoic chamber, with simultaneous recording of sound signals via a microphone. The CART algorithm classified and labeled statistical features derived from the sound signals. Logistic regression and random forest models were then used for failure stage classification, achieving F1 scores of 0.9790 and 0.9220, respectively. Notably, logistic regression achieved the highest score among the classifiers. Conversely, when examining the classifier’s probability change, the probabilities were often skewed toward 0 or 1. This skewed distribution made it difficult to accurately track the forklift’s state changes. Consequently, monitoring the forklift’s degradation status was accomplished through the random forest’s generated classification probabilities. The results showed a gradual change in the condition of the forklift over time. Based on these results, the probabilities generated by the random forest for each stage were combined and averaged to create a degradation curve. The failure point was predicted by the failure threshold that was determined through degradation curve analysis.

In conclusion, predicting the gravity center of objects carried by the forklift yields insights into operations that impact forklift durability. This approach enhances equipment longevity and operational safety. Additionally, providing RUL data facilitates the development of operational plans and efficient maintenance by identifying critical features for setting failure thresholds.

## Figures and Tables

**Figure 2 sensors-23-07706-f002:**
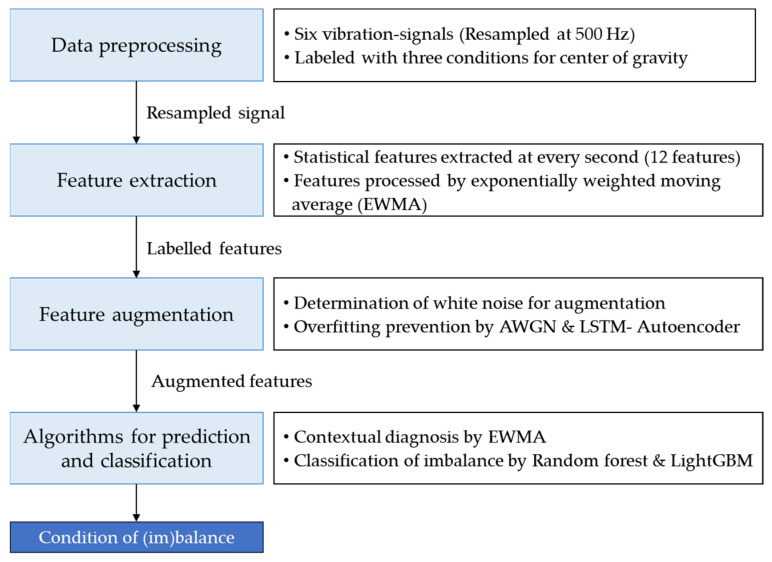
Classification procedure of imbalanced operations based on the center of gravity.

**Figure 3 sensors-23-07706-f003:**
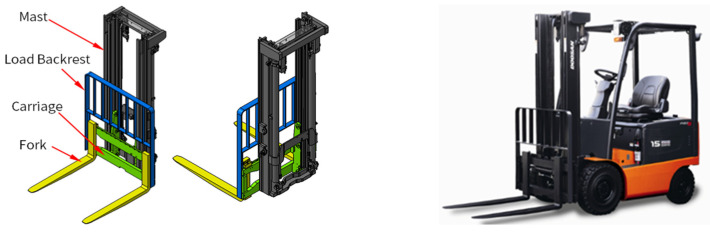
Electric-powered counterbalance forklift of ITA Class I type.

**Figure 4 sensors-23-07706-f004:**
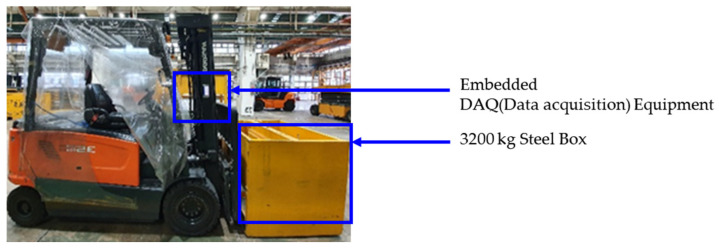
The location of DAQ and the apparatus of a forklift carrying object.

**Figure 5 sensors-23-07706-f005:**
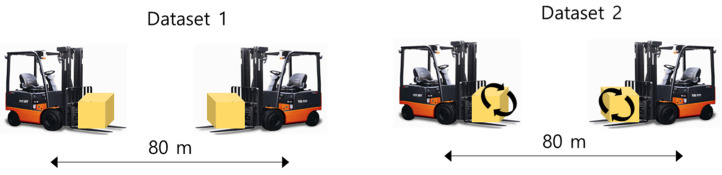
Two operational scenarios for test datasets: driving only (**left**), complex mode (**right**).

**Figure 6 sensors-23-07706-f006:**
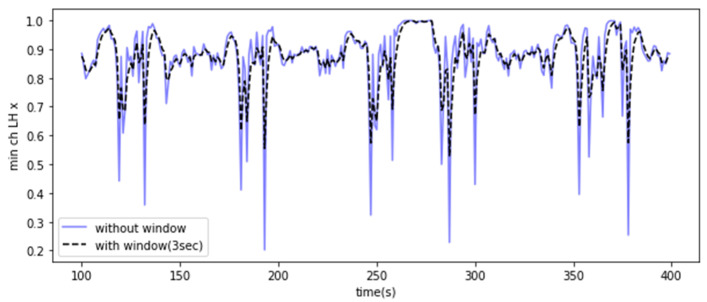
EWMA-processed feature (min) extracted from the x-acceleration signal.

**Figure 11 sensors-23-07706-f011:**
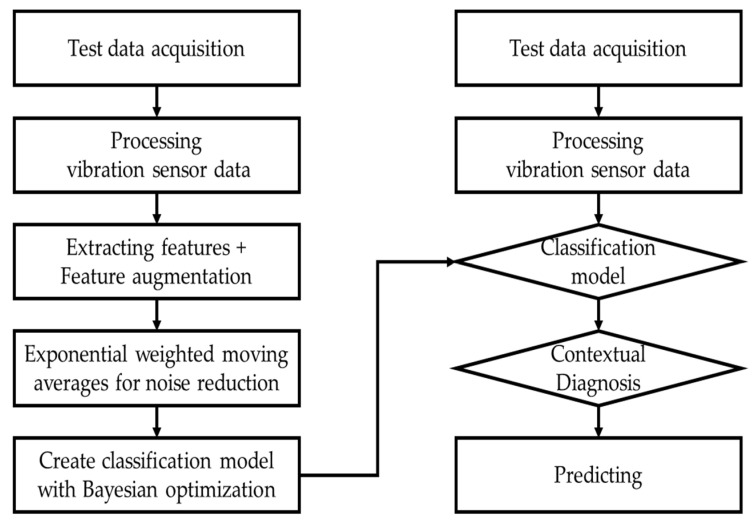
Schematic procedure of training and prediction.

**Figure 12 sensors-23-07706-f012:**
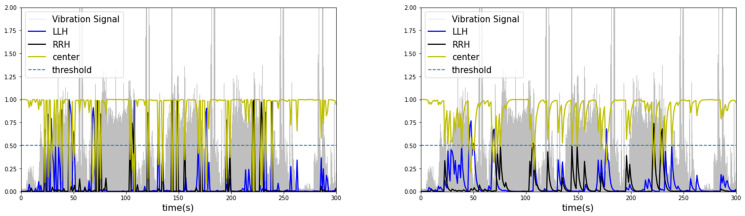
Contextual diagnosis graph of lightGBM (**left**: original, **right**: after moving average).

**Figure 13 sensors-23-07706-f013:**
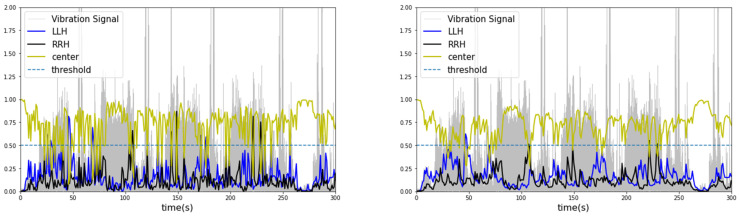
Contextual diagnosis graph of random forest (**left**: original, **right**: after moving average).

**Figure 14 sensors-23-07706-f014:**
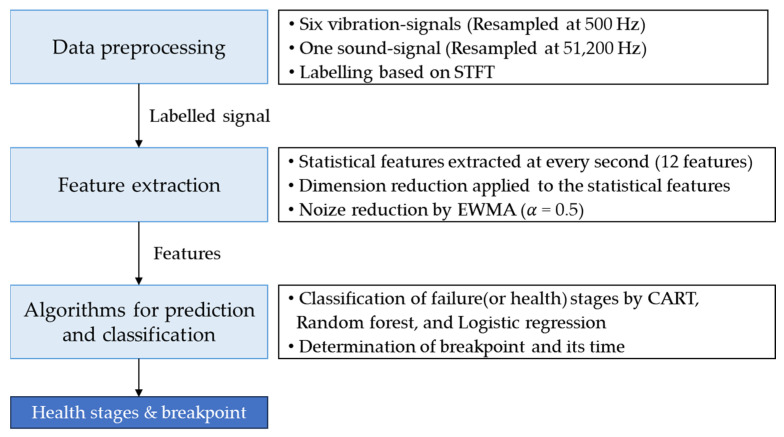
Classification of failure (health) stages in imbalanced abnormal operations.

**Figure 15 sensors-23-07706-f015:**
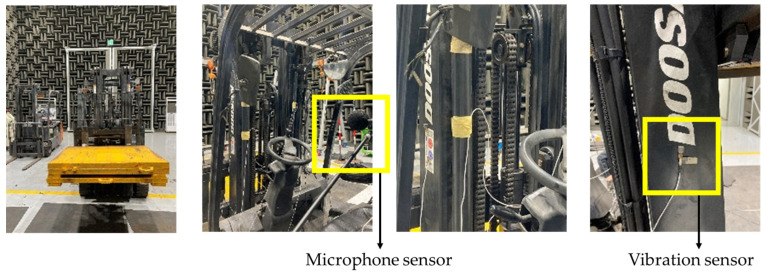
Images of the repeated abnormal lifting experiment in an anechoic chamber with microphone and vibration sensors.

**Figure 16 sensors-23-07706-f016:**
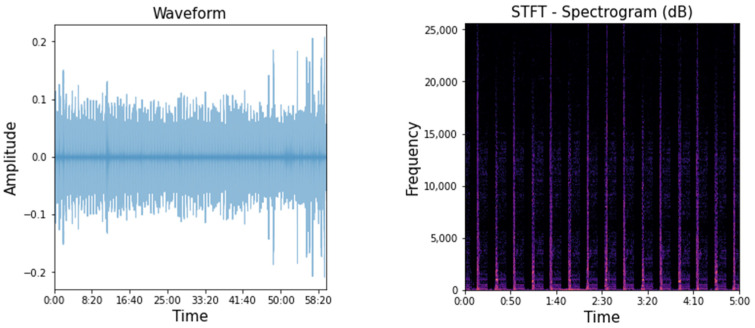
Waveform and short-time Fourier transform (STFT) of measured sound data.

**Figure 17 sensors-23-07706-f017:**
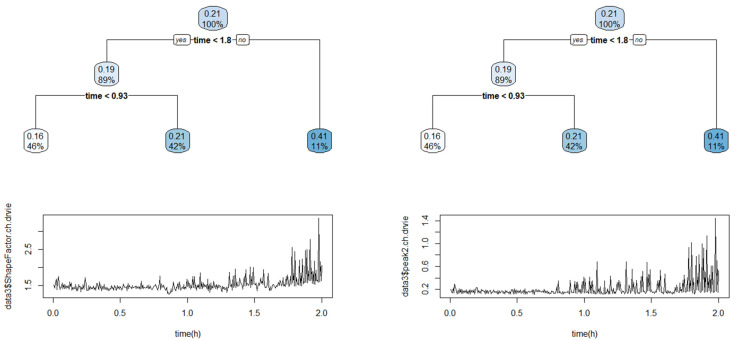
Visual examples of CART algorithm results.

**Figure 18 sensors-23-07706-f018:**
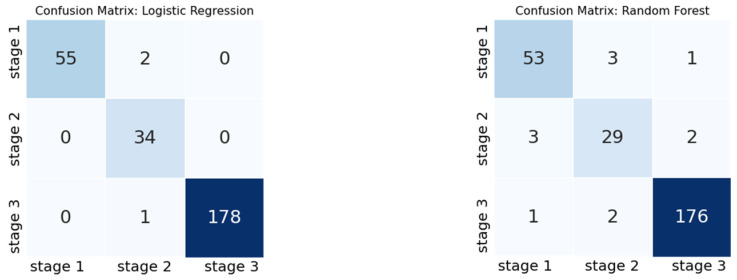
Confusion matrix result: (**left**) logistic regression, (**right**) random forest.

**Figure 19 sensors-23-07706-f019:**
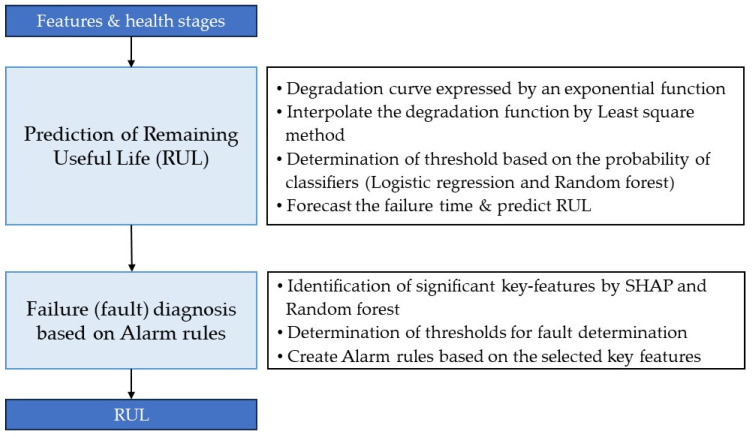
Prediction of RUL and fault diagnosis.

**Figure 20 sensors-23-07706-f020:**
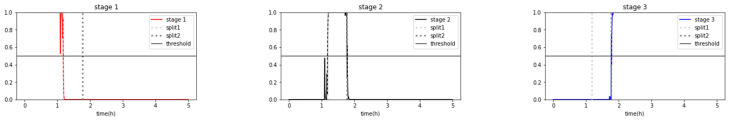
Probability of logistic regression classifier.

**Figure 21 sensors-23-07706-f021:**
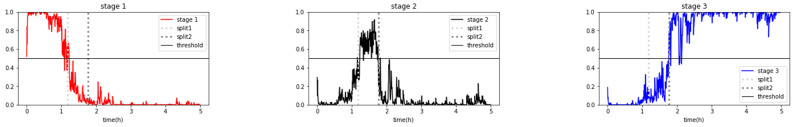
Probability of the random forest classifier.

**Figure 22 sensors-23-07706-f022:**
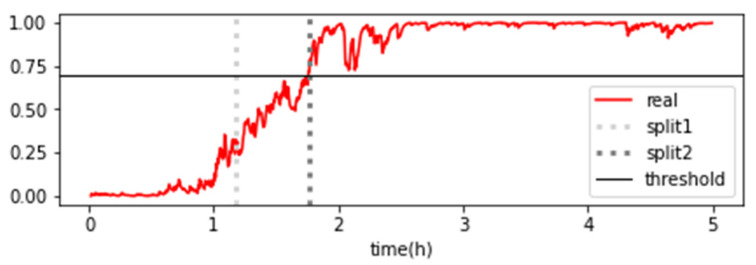
Degradation curve of forklift using the average probability of a random forest classifier.

**Figure 23 sensors-23-07706-f023:**
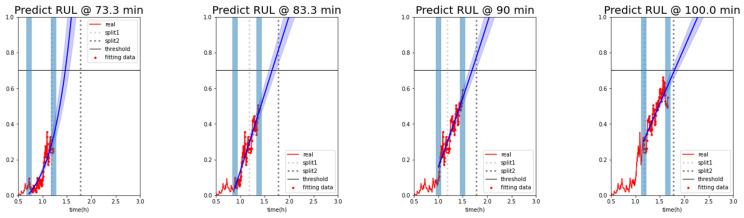
Degradation curve of forklift using the average probability of a random forest classifier.

**Figure 24 sensors-23-07706-f024:**
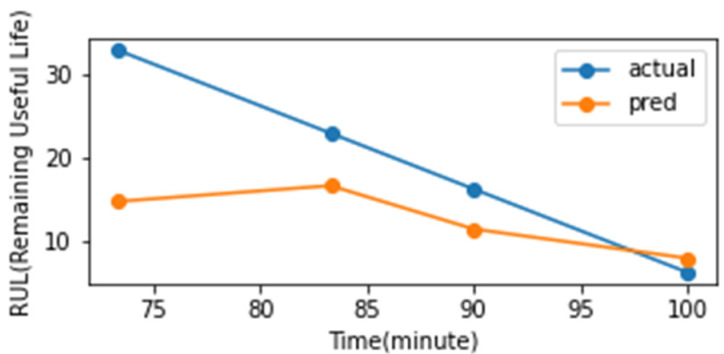
Remaining useful lifetime (RUL) result.

**Figure 25 sensors-23-07706-f025:**
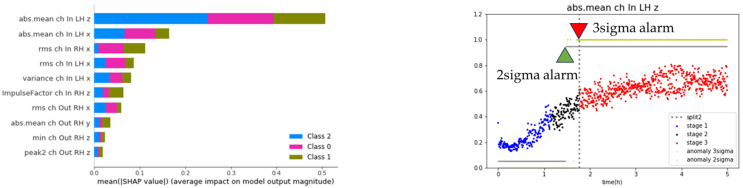
Results of SHAP and fault diagnosis based on the alarm rules.

**Table 1 sensors-23-07706-t001:** Description of features (crest factor, shape factor, impulse factor, and margin factor).

Feature	Dimension
Crest factor	max/RMS
Shape factor	RMS/mean (abs)
Impulse factor	max/mean (abs)
Margin factor	max/mean ((abs)^2^)

**Table 2 sensors-23-07706-t002:** Table of features before applying the exponentially weighted moving average.

Features	Feature Vectors without Exponentially Weighted Window								
	1	…	101	102	103	104	105	…	m
Min	.	…	0.885433	0.843096	0.797650	0.811161	0.827289	…	.
Max	.	…	0.078264	0.080768	0.106396	0.108621	0.116061	…	.
Peak to Peak	.	…	0.093902	0.112324	0.146313	0.142308	0.140491	…	.
Mean (abs)	.	…	0.351697	0.401478	0.407600	0.368202	0.429075	…	.
RMS	.	…	0.267110	0.309412	0.318085	0.291022	0.342365	…	.
Variance	.	…	0.073500	0.098043	0.103476	0.086930	0.119690	…	.
Kurtosis	.	…	0.004885	0.008043	0.011320	0.016485	0.014369	…	.
Skewness	.	…	0.477752	0.469798	0.491220	0.486768	0.492277	…	.
Crest Factor	.	…	0.064416	0.039363	0.096850	0.127109	0.100406	…	.
Shape Factor	.	…	0.049605	0.060854	0.070667	0.080790	0.088413	…	.
Impulse Factor	.	…	0.033882	0.023190	0.053079	0.070003	0.057815	…	.
Margin Factor	.	…	0.003583	0.002052	0.003671	0.005534	0.003556	…	.

**Table 3 sensors-23-07706-t003:** Table of features after applying the exponentially weighted moving average.

Features	Feature Vectors with Exponentially Weighted Window (3 s)								
	1	…	101	102	103	104	105	…	m
Min	.	…	0.875394	0.859245	0.828448	0.819804	0.823547	…	.
Max	.	…	0.076663	0.078716	0.092556	0.100588	0.108325	…	.
Peak to Peak	.	…	0.096910	0.104617	0.125465	0.133887	0.137189	…	.
Mean (abs)	.	…	0.365030	0.383254	0.395427	0.381814	0.405444	…	.
RMS	.	…	0.276765	0.293089	0.305587	0.298304	0.320335	…	.
Variance	.	…	0.079095	0.088569	0.096022	0.091476	0.105583	…	.
Kurtosis	.	…	0.005647	0.006845	0.009082	0.012784	0.013576	…	.
Skewness	.	…	0.479047	0.474423	0.482821	0.484795	0.488536	…	.
Crest Factor	.	…	0.052390	0.045876	0.071363	0.099236	0.099821	…	.
Shape Factor	.	…	0.048446	0.054650	0.062658	0.071724	0.080069	…	.
Impulse Factor	.	…	0.027911	0.025551	0.039315	0.054659	0.056237	…	.
Margin Factor	.	…	0.002978	0.002515	0.003093	0.004314	0.003935	…	.

**Table 4 sensors-23-07706-t004:** Configuration of feature vectors in the dataset.

		Label: Center	Label: Left	Label: Right	Sum

Dataset 1		1244	1209	1246	3699
Dataset 2		1910	1928	1917	5755
Sum		3154	3137	3163	9454

**Table 5 sensors-23-07706-t005:** Dataset size based on feature augmentation.

Feature Combination	No. of Training Data	No. of Test Data
1. Original	6617	2837
2. Original + LSTM AE	13,234	2837
3. Original + AWGN	13,234	2837
4. Original + LSTM AE + AWGN	19,851	2837

**Table 6 sensors-23-07706-t006:** Hyperparameter tuning results of the random forest and lightGBM.

Random Forest		LightGBM	
bootstrap	True	boosting_type	‘gbdt’
ccp_alpha	0	class_weight	None
class_weight	None	colsample_bytree	1
criterion	‘gini’	importance_type	‘split’
max_depth	None	learning_rate	0.02
max_features	‘auto’	max_depth	12
max_leaf_nodes	None	min_child_samples	20
max_samples	None	min_child_weight	0.001
min_impurity_decrease	0	min_split_gain	0
min_impurity_split	None	n_estimators	1000
min_samples_leaf	1	num_leaves	58
min_samples_split	2	random_state	None
min_weight_fraction_leaf	0	reg_alpha	0
n_estimators	130	reg_lambda	0
oob_score	False	silent	−1
random_state	42	subsample	0.8
warm_start	False	subsample_for_bin	200,000

**Table 7 sensors-23-07706-t007:** Result of contextual diagnosis.

Before Contextual Diagnosis				After Contextual Diagnosis			
Time	Left (LLH)	Right (RRH)	Center	Time	Left (LLH)	Right (RRH)	Center
1	0.00112	0.00014	0.99874	1	0.00112	0.00014	0.99874
2	0.00305	0.00044	0.99651	2	0.00241	0.00034	0.99725
3	0.00239	0.00070	0.99691	3	0.00240	0.00055	0.99706
4	0.00103	0.00012	0.99885	4	0.00167	0.00032	0.99801
5	0.00117	0.00022	0.99862	5	0.00141	0.00027	0.99832
6	0.02492	0.00198	0.97311	6	0.01335	0.00113	0.98552
7	0.06715	0.01467	0.91818	7	0.04046	0.00795	0.95158
8	0.06500	0.00303	0.93197	8	0.05278	0.00548	0.94174
9	0.13749	0.01739	0.84512	9	0.09522	0.01145	0.89333
10	0.07009	0.01556	0.91436	10	0.08264	0.01351	0.90385
.	…	…	…	.	…	…	…
.	…	…	…	.	…	…	…
.	…	…	…	.	…	…	…
m	…	…	…	m	…	…	…

**Table 8 sensors-23-07706-t008:** Result of case studies.

Case No.	Dataset	Machine Learning Model	Feature Moving Average Window Size	Smoothing Factor (α)	Contextual Diagnosis Accuracy Score (Probability Moving Average Window Size)		
					1 s	2 s	3 s
Case 1	Raw features	Random forest	1 s	1.00	0.7522	0.8135	0.8950
Case 2	Raw features	Random forest	2 s	0.67	0.8019	0.8622	0.9186
Case 3	Raw features	Random forest	3 s	0.50	0.8347	0.8752	0.9274
Case 4	With LSTM AE features	Random forest	1 s	1.00	0.7392	0.8047	0.8904
Case 5	With LSTM AE features	Random forest	2 s	0.67	0.7846	0.8470	0.9094
Case 6	With LSTM AE features	Random forest	3 s	0.50	0.8216	0.8713	0.9263
Case 7	With AWGN features	Random forest	1 s	1.00	0.7487	0.8238	0.9020
Case 8	With AWGN features	Random forest	2 s	0.67	0.7994	0.8601	0.9203
Case 9	With AWGN features	Random forest	3 s	0.50	0.8294	0.8819	0.9366
Case 10	With all features	Random forest	1 s	1.00	0.7487	0.8587	0.9362
Case 11	With all features	Random forest	2 s	0.67	0.8033	0.8897	0.9450
Case 12	With all features	Random forest	3 s	0.50	0.8305	0.9048	0.9563
Case 13	Raw features	lightGBM	1 s	1.00	0.7659	0.8453	0.9295
Case 14	Raw features	lightGBM	2 s	0.67	0.8223	0.8858	0.9496
Case 15	Raw features	lightGBM	3 s	0.50	0.8646	0.9129	0.9637
Case 16	With LSTM AE features	lightGBM	1 s	1.00	0.7621	0.8347	0.9098
Case 17	With LSTM AE features	lightGBM	2 s	0.67	0.8160	0.8773	0.9369
Case 18	With LSTM AE features	lightGBM	3 s	0.50	0.8488	0.9041	0.9521
Case 19	With AWGN features	lightGBM	1 s	1.00	0.7642	0.8421	0.9161
Case 20	With AWGN features	lightGBM	2 s	0.67	0.8379	0.8925	0.9485
Case 21	With AWGN features	lightGBM	3 s	0.50	0.8643	0.9122	0.9591
Case 22	With all features	lightGBM	1 s	1.00	0.7638	0.8389	0.9221
Case 23	With all features	lightGBM	2 s	0.67	0.8206	0.8883	0.9454
Case 24	With all features	lightGBM	3 s	0.50	0.8569	0.9115	0.9566

**Table 9 sensors-23-07706-t009:** Stage-labeling results of the sound data features using the CART (classification and regression tree) algorithm.

Feature	Breakpoint 1	Breakpoint 2
Max	0.892	1.764
Min	0.925	1.775
Peak2	0.925	1.775
Skewness	1.308	1.831
Crest Factor	1.353	1.708
Shape Factor	1.308	1.775
Impulse Factor	1.353	1.764
Margin Factor	1.353	1.775
Average	1.177	1.771

**Table 10 sensors-23-07706-t010:** Labeling and structure of a dataset.

		Stage 1	Stage 2	Stage 3	Total

Phase		Normal	Semi-failure	Failure	-
Dataset Range (h)		0–1.18	1.18–1.77	1.77–5.0	0–5.0
Number of data		212	106	582	900

**Table 11 sensors-23-07706-t011:** Validation result of stage classification.

Model	Dataset	Accuracy	F1 Score	F1 Score
			(Weighted)	(Macro)
Logistic Regression	Case 1 (vibration feature)	0.9815	0.9814	0.9599
	Case 2 (vibration + sound feature)	0.9889	0.9891	0.9790
Random Forest	Case 1 (vibration feature)	0.9519	0.9523	0.9116
	Case 2 (vibration + sound feature)	0.9556	0.9556	0.9220

## Data Availability

The data presented in this study are available upon request from the corresponding author. The data derived from the present study are only partially available for research purposes.

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
