# Peer review of "Preventing Forklift Front-End Failures: Predicting the Weight Centers of Heavy Objects, Remaining Useful Life Prediction under Abnormal Conditions, and Failure Diagnosis Based on Alarm Rules"

_sensors, 2023, doi:10.3390/s23187706_

Round 1
Reviewer 1 Report
The authors introduce a method of forklift front-end failure prevention including the weight center of heavy objects predictions, RUL prediction, and fault diagnosis. This work is interesting and meaningful in engineering applications. However, the paper needs some necessary revisions before accepting. The detailed comments are as follows:
1. The authors should emphasize more clearly the innovations of their work. After reading the manuscript, it seems that the proposed method is just a combination of existing methods.
2. The content of this paper is rich, including three main parts, i. e. weight center prediction, RUL prediction, and fault diagnosis. However, the readers need to read the related content carefully to understand the relationship between these three parts, so it is recommended to add a systematic diagram.
3. More classical and recent papers about PHM are necessary to supplement the literature review. The following papers could be considered.
Machinery Health Prognostics: A Systematic Review from Data Acquisition to RUL Prediction. 10.1016/j.ymssp.2017.11.016
Lightweight Multiscale Convolutional Networks With Adaptive Pruning for Intelligent Fault Diagnosis of Train Bogie Bearings in Edge Computing Scenarios. 10.1109/TIM.2022.3231325
4. For the introduction of the algorithms, it is recommended to add clearer mathematical descriptions or pseudo-code.
5. The quality of the pictures in the paper needs to be improved, such as in Fig. 14 and Fig 16.
There are some grammatical errors in the manuscript that need to be corrected.
Author Response
Dear Editor and Reviewers,
I trust this message finds you well. We sincerely appreciate the thorough review of our manuscript titled 'Preventing Forklift Front-End Failures: Predicting the Weight Center of Heavy Objects, Remaining Useful Life Prediction under Abnormal Conditions, and Failure Diagnosis based on Alarm Rules'. We extend our gratitude for the invaluable insights provided by the reviewers. We are pleased to submit our revised manuscript in response to their comments.
We are thankful for the constructive feedback from the two reviewers, which has significantly enhanced the quality and rigor of our work. We've taken their suggestions to heart and implemented substantial revisions to improve the clarity, depth, and impact of our paper. The reviewers' recommendations have led us to incorporate additional references and clarify various aspects of the manuscript, ensuring a clearer conveyance of the significance and contributions of our work.
Enclosed, please find the revised manuscript and our rebuttal and response letter. We are confident that the revised manuscript contributes meaningfully to the discourse in the field of Prognostics and Health Management (PHM).
Once again, we appreciate your and the reviewers' insights. We eagerly anticipate hearing from you regarding the outcome of the review process.
Best regards,
Jang Hyun Lee, Ph.D., Prof.
Department of Naval Architecture and Ocean Engineering, INHA University, KOREA
Email: [email protected]

Reviewer 2 Report
This study diagnosed the center of heavy objects carried by the forklift to prevent the failure of the forklift front-end. Overall, the paper is quite interesting. But there are some problems in innovation, literature review, result comparison, language expression and so on. I only agree to the publication of this paper after the authors have revised it according to my comments. Here are some specific suggestions:
1. It is recommended that the authors summarize the contribution of the method proposed in this article in the literature review section of the article to highlight the research advantages of this article
2. The topic of the article is fault diagnosis and RUL prediction, but there is a lack of analysis for specific work in the literature analysis. It is recommended to analyze the recent work of fault diagnosis and RUL prediction. The authors can refer to an integrated multitasking intelligent bearing fault diagnosis scheme based on representation learning under imbalanced sample condition, a parallel hybrid neural network with integration of spatial and temporal features for remaining useful life prediction in prognostics
3. Formula (1-3) lacks sufficient parameter explanation
4. The article uses many methods for analysis, such as AWGN, LSTM autoencoders, random forest and lightGBM models, least squares method, but these methods are not reflected in the method description. As a research paper, there needs to be a core method theory, not a simple method application
5. Regarding the part of RUL prediction, the article only lists the results without comparison
Analysis, it is necessary to compare with some recent work analysis, you can refer to an integrated multi-head dual sparse self-attention network for remaining useful life prediction
6. There are many pictures in the article that are not clear enough, such as Figure 14, 15, etc., the text in the picture is not clear, it is recommended that the authors make corrections
7. There is a problem with the innovation of this article. It is only a list or application of existing methods, lacking core method contributions
Please see the comments to the authors.
Author Response
Response to Reviewer Comments for Manuscript ID [sensors-2585835]
Dear Editor and Reviewer
I trust this message finds you well. We sincerely appreciate the thorough review of our manuscript titled 'Preventing Forklift Front-End Failures: Predicting the Weight Center of Heavy Objects, Remaining Useful Life Prediction under Abnormal Conditions, and Failure Diagnosis based on Alarm Rules'. We extend our gratitude for the invaluable insights provided by the reviewers. We are pleased to submit our revised manuscript in response to their comments.
We are thankful for the constructive feedback from the two reviewers, which has significantly enhanced the quality and rigor of our work. We've taken their suggestions to heart and implemented substantial revisions to improve the clarity, depth, and impact of our paper. The reviewers' recommendations have led us to incorporate additional references and clarify various aspects of the manuscript, ensuring a clearer conveyance of the significance and contributions of our work.
Enclosed, please find the revised manuscript and our rebuttal and response letter. We are confident that the revised manuscript contributes meaningfully to the discourse in the field of Prognostics and Health Management (PHM).
Once again, we appreciate your and the reviewers' insights. We eagerly anticipate hearing from you regarding the outcome of the review process.
Best regards,
Jang Hyun Lee, Ph.D., Prof.
Department of Naval Architecture and Ocean Engineering, INHA University, KOREA
Email: [email protected]

Round 2
Reviewer 2 Report
Thanks to the authors' revision. I accept its publication.
Thanks to the authors' revision. I accept its publication.